# Neural coding in barrel cortex during whisker-guided locomotion

Nicholas James Sofroniew[1†], Yurii A Vlasov[1,2†], Samuel Andrew Hires[1‡], Jeremy Freeman[1], Karel Svoboda[1*]

[1]Janelia Research Campus, Howard Hughes Medical Institute, Ashburn, United States; [2]IBM Thomas J. Watson Research Center, New York, United States

**Abstract** Animals seek out relevant information by moving through a dynamic world, but sensory systems are usually studied under highly constrained and passive conditions that may not probe important dimensions of the neural code. Here, we explored neural coding in the barrel cortex of head-fixed mice that tracked walls with their whiskers in tactile virtual reality. Optogenetic manipulations revealed that barrel cortex plays a role in wall-tracking. Closed-loop optogenetic control of layer 4 neurons can substitute for whisker-object contact to guide behavior resembling wall tracking. We measured neural activity using two-photon calcium imaging and extracellular recordings. Neurons were tuned to the distance between the animal snout and the contralateral wall, with monotonic, unimodal, and multimodal tuning curves. This rich representation of object location in the barrel cortex could not be predicted based on simple stimulus-response relationships involving individual whiskers and likely emerges within cortical circuits.

**\*For correspondence:** svobodak@janelia.hhmi.org

[†]These authors contributed equally to this work

**Present address:** [‡]Biological Sciences, University of Southern California, Los Angeles, United States

**Competing interests:** The author declares that no competing interests exist.

## Introduction

Animals must understand the spatial relationships of objects in their environment for navigation. Rodents move their whiskers to localize (*Knutsen et al., 2006*; *O'Connor et al., 2010*) and identify (*Anjum et al., 2006*) nearby objects, and to guide their locomotion along walls and through narrow tunnels (*Vincent, 1912*; *Sofroniew and Svoboda, 2015*). The neuronal responses to passive deflections of whiskers (*Simons, 1978*; *Armstrong-James et al., 1992*; *Simons et al., 1992*) and whisker-object touches (*Krupa et al., 2004*; *Curtis and Kleinfeld, 2009*; *O'Connor et al., 2010*; *2013*; *Hires et al., 2015*; *Peron et al., 2015*) have been well studied; but little is known about neural coding during natural behaviors, such as tracking a wall during whisker-based navigation. Studying tactile sensation during such natural behaviors is important, because the sensors themselves - digits, whiskers - move in an adaptive manner to produce sensory input.

Touches between whiskers and objects cause deformations of the whiskers and forces at the base of the whisker (*Birdwell et al., 2007*; *Pammer et al., 2013*) that are translated into neural excitation (*Zucker and Welker, 1969*; *Stuttgen et al., 2008*) by mechanosensory receptors (*Dörfl, 1985*). Whiskers are elastic rods with a gradually decreasing thickness as a function of distance from the face (*Birdwell et al., 2007*; *Hires et al., 2013*). Because of the whisker taper, more proximal deflections cause larger forces at the base of the whisker than distal deflections (*Birdwell et al., 2007*; *Pammer et al., 2013*). Objects that are closer to the face therefore trigger higher spike rates in primary sensory neurons (*Szwed et al., 2006*).

Excitation from primary sensory neurons ascends through the brainstem and thalamus to the whisker representation area of the somatosensory cortex, also referred to as barrel cortex (*Van der Loos and Woolsey, 1973*; *Diamond et al., 2008*; *Bosman et al., 2011*). Layer (L) 4 is the main thalamocortical recipient layer and contains spatial clusters of neurons, termed 'barrels'. There is a one-

**eLife digest** Mice are primarily nocturnal animals that rely on their whiskers to navigate dark underground burrows and winding corridors. When a whisker touches an object, cells called neurons at the base of the whiskers produce electrical signals that are relayed to other neurons in an area of the brain called the barrel cortex. However, it is not clear how information is encoded in these electrical signals, in part, because it is technically challenging to collect data about neuron activity and behavior while the mice move around.

To overcome these difficulties, Sofroniew, Vlasov et al. used a touch-based (or 'tactile') virtual reality system to study how mice navigate along corridors. The system simulated the contact the whiskers would have with the walls of a winding corridor. This was achieved by moving the walls with motors while holding the mouse still enough to be able to measure the activity of neurons in the barrel cortex and observe the behavior of the animal.

The experiments show that the electrical signals in the barrel cortex encode information about motion as well as the distance between the mouse and the wall. For example, some neurons in the barrel cortex were only activated when a mouse was a particular distance from the walls. The experiments suggest that the barrel cortex processes signals received from several whiskers to build an overall picture of the locations and shapes of objects.

Sofroniew, Vlasov et al. also used a technique called optogenetics to deliberately activate particular neurons in a manner that mimics their activity patterns during interactions with walls. In the absence of walls, the optogenetic stimuli guided the behavior of the mice so that they tracked along the paths of 'illusory' corridors. Together, these findings reveal the neural code in the barrel cortex that allows mice to navigate by touch.

to-one correspondence between L4 barrels and specific individual whiskers on the face (*Van der Loos and Woolsey, 1973*). Neurons in layers 2/3, 5 & 6, above and below a particular barrel, are part of a 'barrel column'. Neurons in different layers interact through intricate excitatory and inhibitory neural circuits (*Shepherd and Svoboda, 2005*; *Lefort et al., 2009*; *Pouille et al., 2009*; *Hooks et al., 2011*; *Adesnik et al., 2012*; *Pluta et al., 2015*).

Passive whisker deflection causes brief, short latency (~10 ms) excitatory responses in neurons across layers of the corresponding barrel column (*Simons, 1978*; *Armstrong-James et al., 1992*; *De Kock et al., 2007*). Individual neurons are sensitive to the direction (*Simons and Carvell, 1989*), amplitude and velocity of whisker deflection, and neural responses increase monotonically with the strength of a whisker stimulus (*Simons, 1978*). Deflection of neighboring whiskers causes longer-latency, weaker, excitatory responses (*Armstrong-James et al., 1992*) and surround suppression (*Simons, 1985*; *Moore and Nelson, 1998*), implying that neighboring barrel columns can inhibit each other. Similarly, sustained activation of one layer can have mainly inhibitory effects on other layers (*Olsen et al., 2012*; *Pluta et al., 2015*), which could underlie complex responses of barrel cortex neurons that are sometimes seen during active sensation involving prolonged interactions with an object (*Krupa et al., 2004*).

During exploration, mice move their whiskers rhythmically over objects at ~16 Hz (*Mitchinson et al., 2011*; *Sofroniew et al., 2014*; *Sofroniew and Svoboda, 2015*). In a whisker-based pole localization task in head-fixed mice, neurons in layer 4 and layer 2/3 of the barrel cortex show activity highly correlated with touch onset (*O'Connor et al., 2010*; *Crochet et al., 2011*; *Hires et al., 2015*), with larger forces producing more vigorous responses (*O'Connor et al., 2013*; *Peron et al., 2015*). These responses are sensitive to the phase of the whisker at touch (*Curtis and Kleinfeld, 2009*) and the direction of deflection (*Andermann and Moore, 2006*; *Peron et al., 2015*). Rodents can use the identity, magnitude, and timing of whisker-object touch to determine the spatial location of nearby objects (*Brecht et al., 1997*; *Diamond et al., 2008*; *Knutsen and Ahissar, 2009*).

Here, we explored coding in the whisker system during naturalistic behavior. We used a tactile virtual reality system that combines stimulus control and rich behavior with neural recordings and optogenetic manipulation to specifically investigate neural coding of wall position and motion across

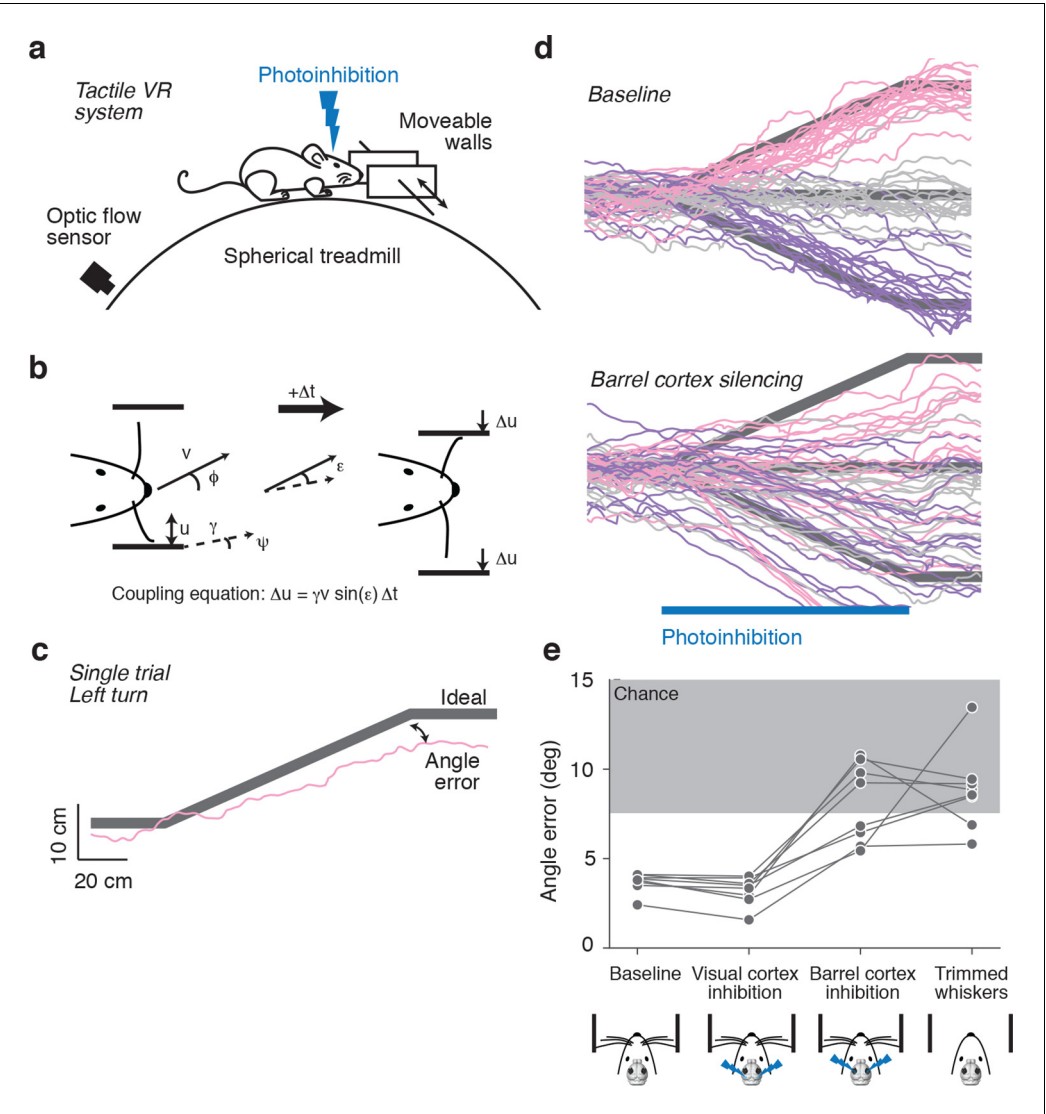

**Figure 1.** Photoinhibition of barrel cortex during wall tracking. (a) Side view of a mouse in the tactile virtual reality system. (b) Schematic illustrating closed-loop control of the walls. If the mouse runs at speed v in the direction φ in a corridor with a turn angle ψ then wall position u updates according to the coupling equation $\Delta u = \gamma\, v\, \sin(\varepsilon)\, \Delta t$ where $\varepsilon = \phi - \psi$ the difference, or error, between the run angle and the turn angle, is the gain, and $\Delta t$ is the time interval. (c) One-left-turn trial. The mouse trajectory (pink) is overlaid on the ideal trajectory (gray), which corresponds to staying in the center of the corridor. The angle error is the difference between the actual and ideal trajectories at the end of the turn. (d) Top, twenty randomly selected running trajectories corresponding to three different turn angles recorded in one session. Bottom, same as above during barrel cortex photoinhibition achieved by photostimulating GABAergic neurons expressing ChR2 (left, pink; straight, gray; right, purple). (e) Average angle error during interleaved trials with no photoinhibition, visual cortex photoinhibition, barrel cortex photoinhibition, and trials with no whiskers. Barrel cortex photoinhibition impaired tracking performance (p = $6.4*10^{-4}$ t-test; 8 mice).

layers of the barrel cortex during whisker-guided locomotion. Our data show how environmental features are transformed and abstracted by somatosensory circuits.

## Results

### Barrel cortex is involved in wall tracking

Mice were head-fixed in a tactile virtual reality system consisting of an air-supported ball and two motorized walls (*Figure 1a*) (*Sofroniew et al., 2014*). The lateral distance between mouse and the walls was controlled in closed-loop with motion of the ball, allowing simulation of a winding corridor. When mice ran towards a wall, a motor moved the wall closer to the animal; when mice ran away from a wall, a motor moved the wall away (*Figure 1b*). When the mice are in the center of the corridor, their whiskers can just touch the walls. Mice use their whiskers to track the walls, staying near the center of the winding corridor without training (*Sofroniew et al., 2014*). This wall-tracking behavior requires mice to interpret forces exerted by the walls on their whiskers in terms of wall distance.

Trials were defined as two-meter-long segments of the corridor that contained either bends to the left or right or were straight. Trials with different bends were randomly interleaved. Mice continued to track the walls and matched their run angle to the bend angle in the corridor (*Figure 1c*). We quantified performance for each trial using angle error, defined as the absolute difference between the run angle of the mouse and the angle of the bend in the corridor.

Whisker-guided locomotion could depend on activity in the barrel cortex, which rodents use in whisker-dependent gap crossing (*Hutson and Masterton, 1986*), object detection (*Miyashita and Feldman, 2013*) and object localization (*O'Connor et al., 2010*). Alternatively, whisker-guided locomotion might be independent of the barrel cortex, for example, relying instead on the superior colliculus, which by itself can mediate orienting behaviors (*Sprague, 1966*) and active avoidance (*Cohen and Castro-Alamancos, 2007*). We tested whether whisker-guided wall tracking uses neural activity in the barrel cortex. Excitatory neurons in the barrel cortex were silenced by photostimulating GABAergic neurons expressing ChannelRhodopsin-2 (ChR2) in VGAT-ChR2-EYFP transgenic mice (*Zhao et al., 2011*). Laser light was focused on the skull, 'photoinhibiting' all cortical layers across a 2-mm-diameter area within 20 ms of laser onset, which was reversed within 150 ms after laser offset (*Guo et al., 2014*). Rapid laser movement between spots on the left and right hemispheres enabled bilateral photoinhibition of the cortex. For silencing barrel cortex, the laser was centered on the C2 barrel, localized using intrinsic signal imaging (*O'Connor et al., 2010*). Mice whiskers were trimmed such that either a single C2 whisker (n = 3) or the C1, C2, and C3 whiskers (n = 5) remained. The corresponding barrel columns fall within the central millimeter of the photoinhibition volume, thus ensuring that the relevant representation in the barrel cortex was abolished by photoinhibition. The task was performed in the dark and stray light from the laser was shielded, eliminating visual cues for locomotion. Photoinhibition lasted for the time needed for mice to run through a one-meter bend (typically 3–7 s).

Silencing barrel cortex bilaterally impaired turning in response to bends in the corridor (angle error 8.09 ± 2.22° vs. 3.68 ± 0.55°, p = $6.4*10^{-4}$ t-test) (*Figure 1d,e*). Performance during photoinhibition was similar to performance after trimming all whiskers (angle error 8.09 ± 2.22° vs. 8.83 ± 2.24°, p = 0.52 t-test). In contrast, there was no effect on turning performance in response to bilateral photoinhibition of the visual cortex (angle error 3.20 ± 0.79° vs. 3.68 ± 0.55°, p = 0.19 t-test), controlling for non-specific effects of photoinhibition. We conclude that mice use neural activity in barrel cortex to guide locomotion along the wall.

### Optogenetic activation of barrel cortex can guide locomotion

Object position can be encoded in spike counts in barrel cortex layer 4 (*O'Connor et al., 2013*). To determine whether activity in the barrel cortex can also drive a wall-tracking like behavior we optogenetically activated layer 4 using *Scnn1a*-TG3-Cre x Ai32 mice. Photoactivation was controlled in a closed-loop with the motion of the ball and in the absence of walls thus creating an 'illusory corridor', where information about distance to illusory walls is encoded in the strength of the photostimulus (*Figure 2a*). Stimuli consisted of 2-ms laser pulses (10 Hz); at maximum experienced power the photostimulus evoked up to 8 additional spikes per neuron per second in layer 4 (*Figure 2—figure supplement 1b*). The power was modulated depending on the position of the mice in the illusory corridor. When mice were in the center of the illusory corridor the laser was off; when mice were on the right side, the left barrel cortex was activated, and vice-versa. The closer the mice were to the

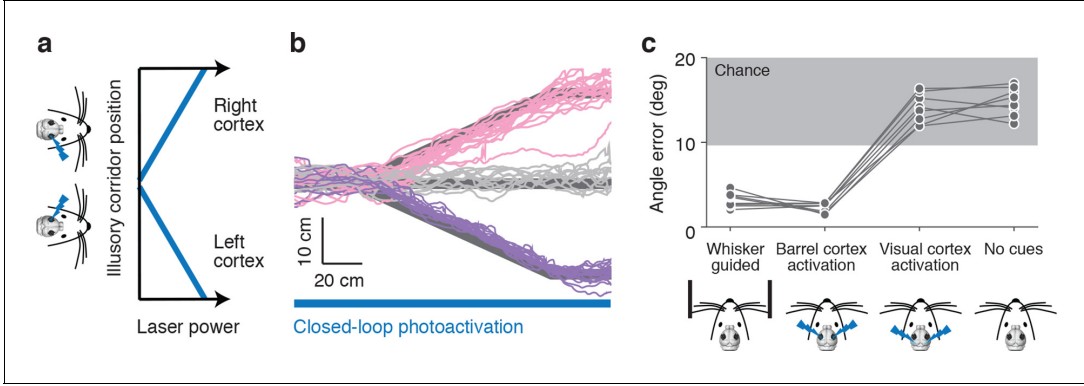

**Figure 2.** Photoactivation of layer 4 to guide locomotion. (**a**) Schematic of an illusory corridor generated with position-dependent photoactivation. In the center of the corridor the laser intensity was zero. On the right side of the corridor the left barrel cortex was stimulated and vice versa. The laser power increased with proximity to the edge of the corridor. (**b**) Twenty randomly selected running trajectories from three different turn angles during closed-loop photoactivation (left, pink; straight, gray; right, purple). (**c**) Average angle error during trials of whisker-based wall-tracking, barrel cortex activation, visual or parietal cortex activation, and no cues. Barrel cortex photoactivation was able to drive a behavior resembling wall tracking (p = $1.7*10^{-8}$ t-test; 8 mice). Trials with barrel cortex activation, visual or parietal cortex activation, and no whisker or photostimulation cues were randomly interleaved. Trials with whisker-based wall-tracking were recorded in separate sessions. Half of the mice performed the photoactivation sessions after the wall-tracking sessions and half of the mice performed them before.

The following figure supplement is available for figure 2:

**Figure supplement 1.** Unilateral activation in mice running in a real corridor.

edge of the illusory corridor, the higher the laser power in the corresponding hemisphere. Mice responded to photoactivation of layer 4 by making left and right turns at bends in the illusory corridor (**Figure 2b**), with slightly lower angle error than when responding to real walls (angle error 2.30 ± 0.85° vs. 3.19 ± 0.55°, p = 0.026 t-test). Running speeds in the illusory corridor were slower (13.2 ± 3.6 cm/s) than running speeds in the corridor with walls (20.4 ± 3.8 cm/s). When parietal (n=3 mice) or visual cortices (n=5 mice) were activated mice did not follow the bends in the corridor (angle error 14.01 ± 1.73° vs. 3.19 ± 0.55°, p = $1.7*10^{-8}$ t-test) (**Figure 2c**). This response to photoactivation was not learnt as all mice performed the task from the very first trial, similarly to how they perform the task with the real walls (**Sofroniew et al., 2014**), and the reward was provided in each trial independent of behavioral performance.

We also unilaterally activated layer 4 with trains of light pulses at constant power as the mice ran through a straight corridor with real walls (**Figure 2—figure supplement 1a**). Running trajectories were biased towards the side of the activation, as if the mice were avoiding a wall they were touching on the contralateral side (7.7 mm ± 4.5 mm at 2.0 mW) (**Figure 2—figure supplement 1c**). Activation of the parietal (n=3) or visual cortex (n=5) did not show an effect on distance from the wall (0.7 mm ± 3.8 mm at 2.0 mW) (**Figure 2—figure supplement 1d**). Together, these results show that activity in the barrel cortex can guide locomotion, and suggest that wall distance may be encoded by spike rates in barrel cortex.

## Barrel cortex contains neurons tuned to wall-distance

To examine population encoding of wall distance, we recorded neuronal activity in layer 2/3 of barrel cortex using 2-photon microscopy and GCaMP6s (**Chen et al., 2013**) in a separate group of mice running through the winding corridor (6 mice). Mice were left with all their whiskers, so we could look at neural coding under their natural whisker configuration. Three 600 x 600 µm planes separated by 20 µm in depth were imaged at 7.1 Hz around the center of the C2 barrel column, found using intrinsic signal imaging (**Peron et al., 2015**). To measure repeated responses across a range of wall distances, we also included 'open-loop' trials in the winding corridor, in which the contralateral wall was moved to fixed distances away from the mouse independent of the animal's locomotion (**Figure 3a**).

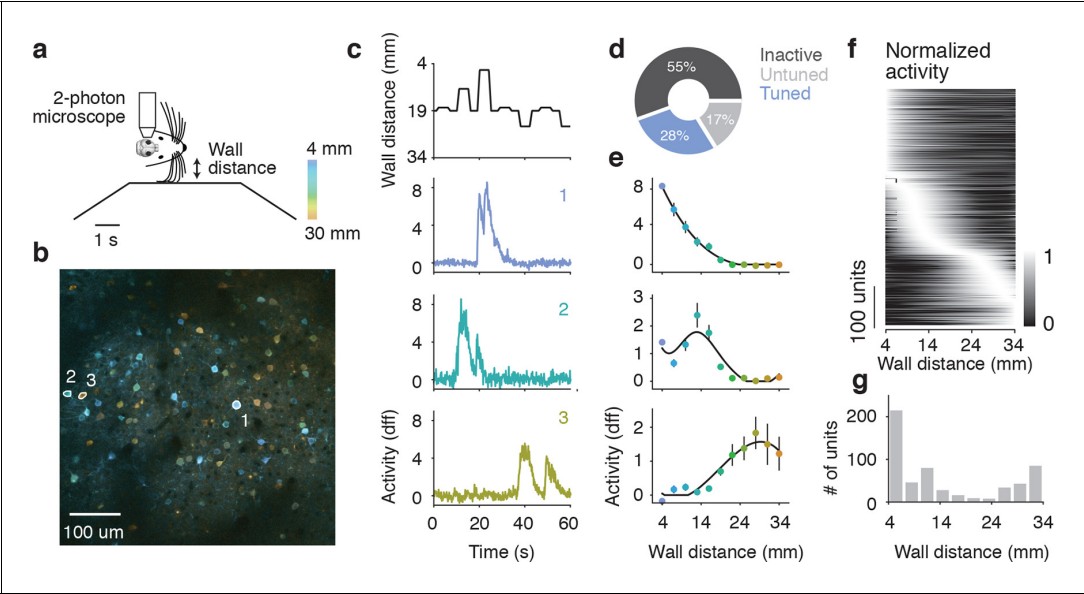

**Figure 3.** Imaging activity of layer 2/3 neurons during whisker-guided locomotion. (**a**) Schematic of two-photon calcium imaging. (**b**) Overlay of a pixelwise regression map and mean intensity image. Each pixel in the regression map is colored according to its tuning to wall distance; brightness was adjusted according to the $r^2$ value of the tuning. Three example ROIs are highlighted (corresponding to panel c). This imaging region is approximately centered on the C2 barrel (diameter, 300 μm) and contains parts of the neighboring D1 and C1 barrels. (**c**) Distance from the snout to the wall as a function of time (top) and $\Delta F/F$ for three example ROIs (same ROIs as in **b**). (**d**) Fraction of neurons in L2/3 that are inactive, tuned, and untuned to the wall distance. (**e**) Tuning curves to wall distance for example ROIs (mean ± SE over trials). (**f**) Heatmap of tuning curves normalized by maximum activity across all mice. (**g**) Histogram of the location of tuning curve peaks.

We investigated the relationship between neural activity and wall distance using two complimentary methods. First, a pixelwise regression (*Ohki et al., 2005*; *Freeman et al., 2014*) related fluorescence changes at each pixel to wall distance. Analysis was restricted to periods of running (speed > 3 cm/s), which ensures that mice were actively whisking and that their whiskers were interacting with the walls (*Sofroniew et al., 2014*). This analysis yielded a map with each pixel colored based on the wall distance at which the pixel was maximally active, and its brightness is based on an $r^2$ value indicating the reliability of that response (*Figure 3b*). Individual neurons appeared as tuned to different wall distances. In a second analysis, we manually defined regions of interest (ROIs) around individual neurons (2019 neurons, 6 mice). Calcium fluorescence time courses from individual ROIs showed tuning to wall distance consistent with the pixel-wise maps (*Figure 3c,e*). For most ROIs, responses increased with decreasing wall distance. For other ROIs, responses peaked at particular wall distances, or even increased with increasing wall distance (*Figure 3f*).

Overall 564/2019 (28%) of ROIs showed significant tuning to wall distance (p<0.05, ANOVA across trials) (*Figure 3d*). A further 334/2019 (17%) of ROIs were active (90th percentile F/F > 1.0) but were not tuned to wall distance, and the remaining 1121/2019 (55%) ROIs were inactive. The peak wall distance preferred by the tuned ROIs tiled the length of the whiskers, although there was a bias towards small distances (*Figure 3f,g*). These imaging experiments revealed a rich representation of wall distance in the superficial layers of the barrel cortex.

## Electrophysiology in barrel cortex during corridor tracking

To examine other cortical layers, and to verify the response properties observed with calcium imaging, we made extracellular recordings from barrel cortex in a separate group of mice running through the winding corridor (*Figure 4a*) (13 mice). We used 32-channel single shank silicon probes in mice expressing ChR2-eYFP in layer 4 neurons (*Scnn1a*-Tg3-Cre x Ai32). We targeted recordings to barrel field by visualizing the eYFP fluorescence during surgery. The probe was coated with DiI, allowing us to reconstruct the probe location within the barrel cortex (*Figure 4b*). Recordings were clustered around the C2 barrel (*Figure 4—figure supplement 1a*). Current source density analysis

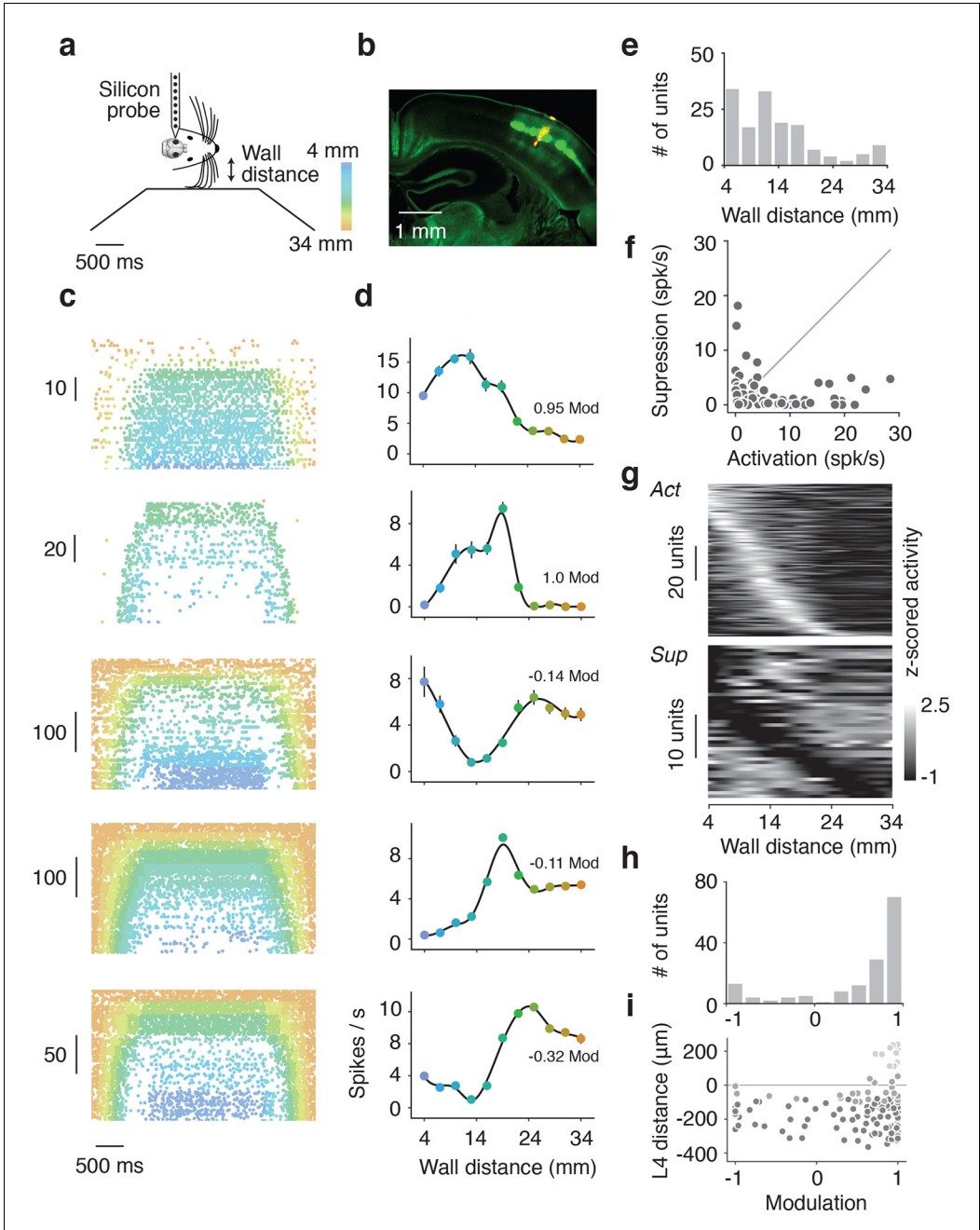

**Figure 4.** Extracellular electrophysiology during whisker-guided locomotion. (a) Schematic showing silicon probe recordings during open loop trials. The wall was moved in and out of different fixed distances from the mouse during a period of 4 s. (b) Coronal section through the brain of an *Scnn1a-Tg3-Cre x RCL-ChR2-EYFP* mouse acquired with green filter showing the barrels. Image acquired with orange filter is superimposed on top showing the track of the silicon probe coated in DiI. Electrolytic lesion is seen at the end of the DiI trace. Location of the probe is identified as C3 barrel. (c) Example spike rasters of regular spiking units during open-loop trials. Each row of the raster corresponds to one trial and each dot corresponds to one spike. The color of each dot represents the position of the wall at the time of the spike. Recordings are performed around C1 and C2 barrels. Only trials with running speed over 3 cm/s are represented. (d) Corresponding tuning curves to wall distance for the spike rasters shown in B (mean ± SE over trials). (e) Histogram of the location of tuning curve peaks. (f) Scatter plot of tuning curve suppression vs. activation. Activation is the difference between peak rate and baseline rate when the wall is out of reach. Suppression is the difference between minimum rate and baseline rate. (g) Heatmaps of z-scored tuning curves for units activated by more than 1 Hz (top) and units suppressed by more than 1 Hz (bottom) sorted by the location of the maximum and minimum wall distances respectively. Units that were both activated and suppressed appear in both plots. (h) Histogram of the tuning modulation, defined as the ratio of the difference between the activation and suppression divided by the sum of the activation and suppression. Units that are just

*Figure 4 continued on next page*

*Figure 4 continued*

activated have modulation 1, units that are just suppressed have modulation −1, and units that are both activated and suppressed have a modulation near 0. (i) Modulation index as a function of laminar position. Light gray, units classified as layer 2/3; medium gray, layer 4; dark gray, layer 5.

The following figure supplements are available for figure 4:

**Figure supplement 1.** Electrophysiological methods.
**Figure supplement 2.** Lamina distribution of units.
**Figure supplement 3.** Comparison of open-loop and closed-loop tuning curves.
**Figure supplement 4.** Effects of running speed on activity and wall distance tuning.
**Figure supplement 5.** Tuning to ipsilateral and contralateral wall distance.

triggered on layer 4 activation of ChR2 and electrolytic lesions were used to calibrate the laminar location of the probe and determine the depth of the recorded neurons (*Figure 4—figure supplement 1b*). Recordings spanned layer 2 to layer 5, although the majority of units were located in infragranular layers (*Figure 4—figure supplement 2a*). Baseline and peak spike rates were higher in deeper layer neurons (*Figure 4—figure supplement 2b,c*). Spike sorting was performed manually, and both false alarm rate of the inter-spike-interval distribution < 1.5% and waveform SNR > 6 were used as quality control metrics (*Figure 4—figure supplement 1c,f*). As with the imaging experiments, analysis was restricted to periods of running (speed > 3 cm/s), ensuring the mice were actively whisking (*Sofroniew et al., 2014*).

We isolated a total of 209 units (13 mice). Of these units, 179 (86%) showed significant tuning to wall distance during open-loop movements of the wall (p<0.05 ANOVA). 148 were classified as regular spiking and presumed excitatory based on spike width (*Figure 4—figure supplement 1d,e*) (*Guo et al., 2014*) and used for further analysis. Individual units showed diverse responses to interactions with the wall that were reproducible across trials (*Figure 4c,d*). Some units had low activity when the wall was out of reach and increased activity monotonically as the wall moved closer or had maximal responses at particular wall distances (*Figure 4c,d*). Other units had high activity when the wall was out of reach and decreased activity monotonically as the wall approached certain wall distances. We also observed more complex responses, including a mix of enhancement and suppression across wall distances. Distances eliciting peak responses tiled the relevant space of wall distance (*Figure 4e*).

For each unit, we quantified activation and suppression as a function of wall distance (*Figure 4f*). The baseline rate of each unit was the spike rate during periods of running when the wall was out of reach of the whiskers. Activation was the difference between peak spike rate and baseline spike rate; suppression was the difference between baseline spike rate and minimum spike rate. Both the peaks and troughs of the activated and suppressed units tiled the relevant wall distance range (*Figure 4g*). A modulation index was calculated as the difference between activation and suppression normalized by the sum of activation and suppression (*Figure 4h*). This index ranged from one, for activated units, to minus one, for suppressed units, with mixed units that are both activated and suppressed near zero. The majority of units showed activation (106/148; 72% modulation > 0.5), but a sizable fraction showed suppression or mixed responses (42/148; 28% modulation <0.5). Units with negative and near-zero modulation indices tended to be found in infragranular layers (*Figure 4i*).

These tuning curves were generated during open-loop movements of the wall, but during natural behavior the wall will move in closed-loop with locomotion. To investigate whether tuning under these conditions is similar, we characterized wall distance tuning during closed-loop wall-movements for a subset of mice (10 mice; 114 regular spiking units). In closed-loop, wall distance sampling is non-uniform because the sensory stimulus depends on behavior. We combined epochs across several widths and bends in the corridor to approximately sample wall distance as in the open-loop condition. Units showed similar tuning curves under both open-loop and closed-loop (*Figure 4—figure supplement 3a,b*), with similar average rates over the same wall distances range (107/114; 94% -0.5

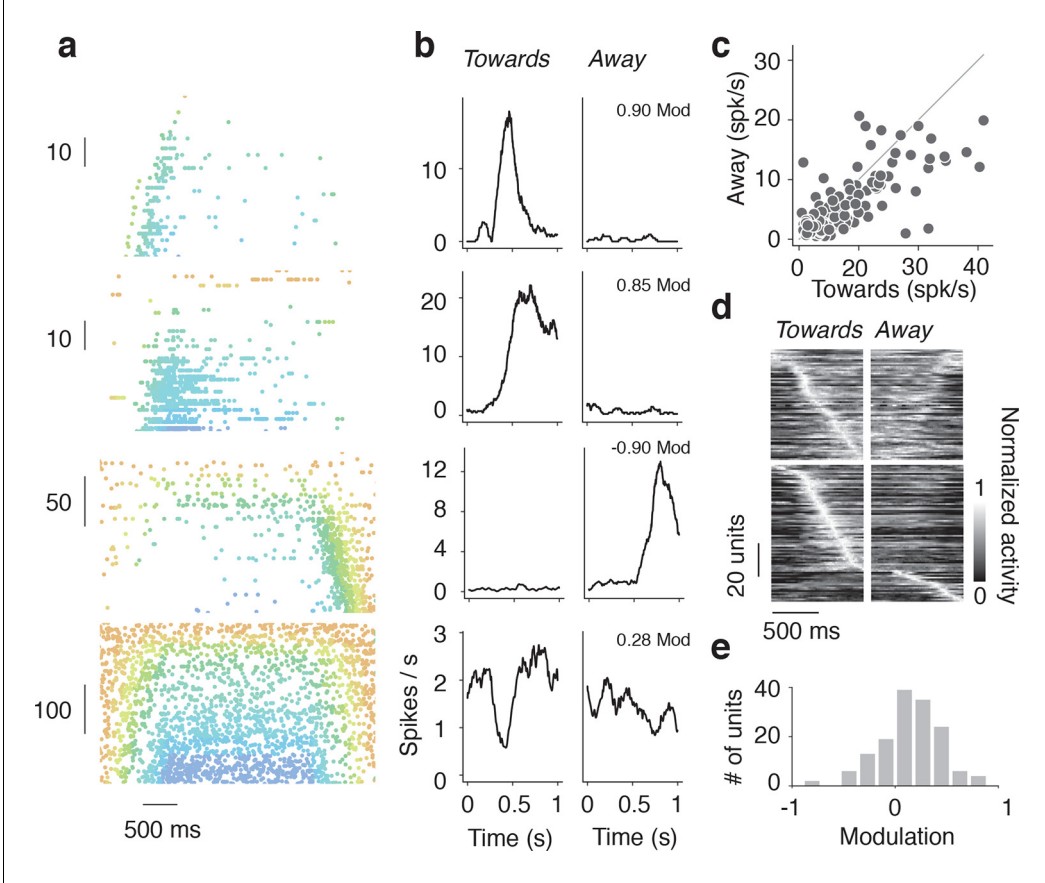

**Figure 5.** Tuning to direction of wall movement. (**a**) Example spike rasters of regular spiking units that showed strong modulation by wall direction during open-loop trials during locomotion (running speed over 3 cm/s). (**b**) Spike rate as a function of time during epochs when wall is moving towards/away from the mice for the same regular spiking units as in **a**. (**c**) Heatmap of time profiles curves normalized by maximum for symmetric and asymmetric units, sorted by time to peak. (**d**) Scatter of the range of spike rates as the wall moved towards and away from the mouse. The range of spike rates is the difference between the maximum and minimum rate during the 1 s when the wall was moving towards or away from the mouse. (**e**) Histogram of direction modulation index, which is the range difference in spike rates during wall movement towards and away from the mouse divided by the sum of the towards range and away range. Units that respond only when the wall approaches have modulation 1, units that respond only when the wall moves away have modulation −1, and units that respond to both the wall approaching and moving away have a modulation near 0.

< open vs. closed modulation < 0.5) (*Figure 4—figure supplement 3c,d*). A substantial fraction of all units (31%; 46/148) were significantly modulated by running speed in the absence of the walls (*Figure 4—figure supplement 4a,b*) (*Keller et al., 2012*; *Saleem et al., 2013*). However, the wall distance tuning curves computed during slow running and fast running were similar on average (modulation index was −0.10 ± 0.55) (*Figure 4—figure supplement 4c,d,e*).

When running in a narrow corridor, whiskers can interact with both walls simultaneously. Information from the whiskers projects directly to the contralateral barrel cortex from subcortical structures; however, a small callosal projection also carries information from one side of the barrel cortex to the other (*Czeiger and White, 1993*). For a subset of mice (3 mice; 29 regular spiking units) we recorded activity in response to open-loop movements of the ipsilateral wall. Compared to interactions with the contralateral wall, very few units were strongly modulated by interactions with the ipsilateral wall (*Figure 4—figure supplement 5a,b,c,d*). Spike rate changes in response to interactions with the contralateral wall were consistently greater than to the ipsilateral wall (27/29; 93% contra vs ipsi modulation > 0) (*Figure 4—figure supplement 5e,f*). This finding is consistent with the results

of the unilateral activation experiment, which suggest the hemisphere contralateral to the wall is predominantly involved in wall-tracking.

Some units exhibited sensitivity not only to wall position, but also to the direction of wall motion. We compared responses when the wall moved towards or away from the mouse on open-loop trials. Some units were activated when the wall distance decreased (*Figure 5a,b*, top two), whereas other units were activated when wall distance increased (*Figure 5a,b*, bottom two). The peak responses of different units tiled the range of distances covered the wall moved either towards or away from the mouse (*Figure 5c*). A direction modulation index was computed by comparing spike rates during movement towards and away (*Figure 5d,e*). Some neurons were strongly modulated (82/148, 55% absolute direction modulation > 0.2). Barrel cortex thus encodes both wall position and direction of wall motion.

## Discussion

We measured neural coding in the barrel cortex during naturalistic wall tracking. Mice used their whiskers to navigate winding corridors in tactile virtual reality (*Sofroniew et al., 2014*). Neural activity in the barrel cortex was necessary for guiding locomotion based on whisker cues (*Figure 1*). In addition, graded activation of layer 4 neurons in the barrel cortex was sufficient to guide locomotion in an illusory corridor (*Figure 2*). Remarkably, similar to closed-loop wall tracking (*Sofroniew et al., 2014*), no training was required for mice to run in an illusory corridor. The tracking accuracy in the illusory corridor was on par or better compared to the real corridor; this may be because the photo-stimulus provided a stronger distance-dependent signal compared to touch-evoked activity. These experiments show that downstream layers and brain areas can interpret increases of layer 4 spike rate as increased proximity to a wall on the contralateral side and drive the appropriate turning responses. Previous experiments suggest key roles for the superior colliculus in orienting and goal-directed behaviors (*Sprague, 1966*; *Cohen and Castro-Alamancos, 2007*). Barrel cortex activity could drive whisker-guided locomotion through the superior colliculus and rubrospinal circuits. These brain areas could also be capable of mediating this behavior independently after chronic barrel cortex lesions.

The optogenetic experiments (*Figure 2*) suggest a simple code for wall distance: wall distance could be coded as inversely proportional to neural activity in layer 4. Sensory neurons in the trigeminal ganglion have monotonic responses to object distance (*Szwed et al., 2006*). Similarly, during passive stimulation (*Simons, 1978*) or simple single-whisker behaviors (*Simons, 1978*; *Peron et al., 2015*) barrel cortex neurons show monotonic relationships with stimulus strength. The majority of L2/3 and L4 (*Figure 4—figure supplement 2b,c*) neurons responded weakly when the wall was out of reach and only responded strongly when the contralateral wall came within reach. However, many neurons exhibited a rich representation of wall distance (*Figure 4h*). Some neurons had non-monotonic unimodal relationships with wall distance and were most active at intermediate wall distances. Across the population, neuronal tuning tiled all wall distances (*Figure 4g*). Activity of other neurons showed even more complex, multimodal tuning to wall distance. Complex responses to wall distance were especially pronounced in L5 (*Figure 4i*). Encoding wall distance with a population of tuning curves that increase monotonically with decreasing wall distance (i.e. sigmoids) would lead to very large metabolic costs when representing small distances. Given a fixed metabolic cost, a population of non-monotonic (i.e. Gaussian) tuning curves that tile the most frequently encountered distances can achieve higher overall encoding accuracy (*Ganguli and Simoncelli, 2014*). Some of the features of these complex tuning curves may also be used for processing by downstream neurons.

We measured tuning curves from layer 2/3 neurons using both calcium imaging (*Figure 3*) and electrophysiology (*Figure 4*). Both recording methods revealed neurons that were strongly activated when the wall approached. The calcium imaging dataset in addition contained some neurons that were activated by running and suppressed monotonically as the wall came closer to the face (*Figure 3c*). These neurons may have been activated by tactile stimuli from the D and E row whiskers that are capable of touching the ball during running, and suppressed as the shorter surround whiskers touched the wall at smaller wall distances. In contrast, in the electrophysiology data the majority of the units were located in the C row barrels and had low baseline activity even during running.

Calcium imaging revealed 28% of neurons as tuned to wall distance. Using similar methods, 17% of neurons responded to touch during a pole location discrimination task performed with a single whisker (*Peron et al., 2015*). These differences could arise from the multi-whisker nature of the wall-tracking experiments, the greater complexity of the interactions between the whiskers and the wall compared to the pole, or the fact that mice were running during wall-tracking.

Neurons were also sensitive to the direction of wall movement. This sensitivity could be produced by neurons that have tuning to the direction of whisker deflection (*Andermann and Moore, 2006*). Wall direction selective neurons could be useful for helping the mice determine wall velocity, and acceleration. Coding for these variables, in addition to wall distance, could help the mice better track the walls through control laws that include derivative feedback alongside proportional feedback (*Sofroniew et al., 2014*).

Multiple mechanisms could contribute to complex, non-monotonic tuning curves in barrel cortex neurons. Mice may have adjusted their whisker movements in subtle ways with wall distance (*Voigts et al., 2015*) which could change the strengths of the interactions with the wall and thus shape tuning curves. The similarity of tuning curves under open-loop and closed-loop conditions (*Figure 4—figure supplement 3*), where running direction are different, suggests that the details of whisker motion are unlikely to be a critical factor. Another mechanism is based on cross-columnar inhibition mediated by multi-whisker interactions (*Simons, 1985*; *Brumberg et al., 1996*; *Moore and Nelson, 1998*). Individual whiskers have different lengths (*Ibrahim and Wright, 1975*) and touch the wall at different times as it approaches. Neurons driven by the long whiskers may be suppressed by neurons driven by the shorter whiskers at smaller wall distances, resulting in non-monotonic distance tuning curves. A third mechanism relies on cross-laminar excitation and inhibition. Neurons in different layers are coupled through interlaminar connections which excite both excitatory neurons and inhibitory interneurons in the target layers (*Lefort et al., 2009*; *Hooks et al., 2011*; *Feldmeyer et al., 2013*; *Pluta et al., 2015*). The balance of excitation and inhibition can change over time scales of milliseconds to seconds. For example, over short time scales the synchronous excitation of L4 neurons (*O'Connor et al., 2013*; *Hires et al., 2015*) is expected to drive spikes in L5 neurons (*O'Connor et al., 2013*). Over longer time scales L4 activation promotes mostly inhibition in L5 (*Pluta et al., 2015*). Cross-columnar and cross-laminar inhibition combined with short-term synaptic plasticity, likely help generate the complex representation of distance in the barrel cortex, which in turn supports wall-tracking behavior.

## Materials and methods

### Mice

Twelve VGAT-ChR2-EYFP BAC (line 8) transgenic mice (Jackson Labs, Bar Harbor, ME: *014548*; *VGAT* corresponds to *Slc321a*) were used for photoinhibition experiments (*Zhao et al., 2011*; *Guo et al., 2014*). Eight *Scnn1a*-TG3-Cre x Ai32 mice were used for photoactivation experiments. *Scnn1a*-TG3-Cre mice (Jackson Labs: *009613*) have Cre expression restricted to ~85% layer 4 excitatory neurons in the cortex and some thalamic neurons (*Madisen et al., 2010*; *Pluta et al., 2015*). Ai32 (Jackson Labs: *012569*) mice contain the Channelrhodopsin-2 (ChR2H134R-EYFP) gene in a Cre-dependent reporter cassette at the Rosa26 locus (*Madisen et al., 2012*). Expression of ChR2 was confirmed in barrel cortex and visual cortex by histology, although expression in parietal cortex is much weaker. Thirteen *Scnn1a*-TG3-Cre x Ai32 mice were used for the electrophysiology. Six C57BL/6Crl (Jackson Labs: *000664*) mice were used for the calcium imaging experiments.

Mice were housed individually in cages with bedding and running wheels (Bio-Serv, Flemington, NJ: K3327 and K3251) in a reverse light-cycle room. Mice were restricted to consume 1.0-1.5 ml of water per day (*Guo et al., 2014*), which could either be obtained during behavioral sessions or in supplements after behavioral sessions. The weight change during the behavioral session was used to estimate the amount of water consumed during the session, and the supplement was chosen accordingly. The weight and health (posture, quality of fur, and motor activity) of the mice were monitored daily. All procedures were in accordance with protocols approved by the Janelia Farm Institutional Animal Care and Use Committee.

## Polished dental cement preparation

For optogenetic experiments, mice were prepared with a clear skull implant (*Guo et al., 2014*). Mice were implanted with headposts. Before starting photostimulation experiments (typically 1-8 weeks after headpost implantation) the surface of the clear dental acrylic was polished (Acrylic Polishing Kit HP Shank, Pearson Dental, Sylmar, CA) and covered with a thin layer of clear nail polish (Electron Microscopy Sciences, Hatfield, PA, 72180). This preparation transmits ~60% of incident light to the brain (*Guo et al., 2014*).

## Laser photostimulation system

Photostimulation of Channelrhodopsin-2 was achieved using a 473 nm DPSS laser (Ultralasers, Toronto, Canada, DHOM-T-473-200). Laser power was controlled using an acousto-optic modulator (AOM) (Quanta Tech, Shoreline, WA, MTS110-A3-VIS) and fixed RF frequency driver (Quanta Tech, MODA110-D4500-2460). The AOM was controlled with a 0-5 V analogue signal from the behavioral control system. The output of the AOM was coupled to a 62.5 μm multimode fibre (Thorlabs Newton, NJ, M31L03) with an FC/PC adaptor (Thorlabs, PAF-X-5-A) and brought inside the light tight box. The light was directed onto a 2D scanning galvo system (Thorlabs, GVSM002). The position of the scan mirrors was controlled with +/− 10 V analogue signals from the behavioral control system. The beam was then expanded 5x with plano-convex lenses (Thorlabs, LA1951-A, and LA1384-A) and focused onto the brain surface with a f = 200 mm lens (Thorlabs, AC508-200-A). The beam diameter on the skull in a system of identical design was 400 μm (*Guo et al., 2014*). The laser path was shielded with a 2"-diameter lens tubes. To ensure complete shielding of the stray laser light for silencing experiments, 3D-printed black plastic pieces were secured to the skull of the mouse and to a lens tube via a black bellows junction (McMaster Carr, Elmhurst, IL, 94205K77). A stereomicroscope (Nikon, Tokyo, Japan, SMZ745, with C-W 10xB [F.N. 22] eyepieces and a G-AL0.5X auxiliary objective) was used to align the laser with vasculature or Bregma.

## Photoinhibition parameters

Bilateral photoinhibition was achieved by deflecting the laser beam between the left and right hemispheres at 100 Hz. On each hemisphere the laser either dwelled at a single spot for 10 ms or dithered between 5 spots arranged in a cross with 300 μm between the spots. For bilateral silencing, three mice were stimulated with average power 25 mW per hemisphere in a 300 μm cross, two mice were stimulated with average power 15 mW per hemisphere in a 300 μm cross, and three mice were stimulated with average power 20 mW per hemisphere at a single spot. For each animal, stimulation of S1 and V1 used the same parameters. Unilateral photoinhibition was achieved by deflecting the laser between the target hemisphere and a spot outside of the preparation at 100 Hz. Four mice were silenced at a single spot with average power 7.5 mW per hemisphere. These parameters result in robust silencing throughout the depth of the cortex over a 2–4 mm$^2$ area (*Guo et al., 2014*). Silencing lasted for the duration of the 100-cm test period, which was generally 2–6 s. Trials silencing S1 and V1 were interleaved randomly with trials without silencing, such that trials of different types occurred with equal probability. Based on intrinsic signal imaging, the coordinates for the C2 barrel were 1.0 mm posterior to Bregma and ± 3.0 mm lateral. Coordinates for V1 were 3.0 mm posterior to Bregma and ± 1.0 mm lateral based on the Allen Brain Atlas (www.brain-map.org). For silencing, mice had either all their whiskers or a single whisker/row of whiskers on each side

## Photoactivation parameters

Layer 4 activation was done at a single spot at 10 Hz with 2-ms pulses. L4 photoactivation rapidly (<10 ms) drives spikes in other layers of the barrel cortex (*O'Connor et al., 2013*). For open-loop biasing experiments peak laser power ranged from 0 to 2 mW. The effects of photoactivation were calibrated using cell-attached recordings in awake but non-running mice (*Figure 2—figure supplement 1b*). For closed-loop activation experiments, peak laser power was scaled to position in the virtual corridor such that the laser was off when mice were at the center of the corridor and at 4.5 mW power if mice were at the edge of the virtual corridor. Mice mainly stayed in the portion of the corridor where the laser power was under 2 mW. The left hemisphere was stimulated if the mouse was on the right side of the virtual corridor, and the right hemisphere was stimulated if the mouse was

on the left side of the virtual corridor. For both open- and closed-loop experiments, stimulation only occurred when the mouse was running. For activation experiments mice had all their whiskers.

Calibrations of the L4 photoactivation were performed in awake *Scnn1a*-TG3-Cre x Ai32 mice. On the day of the recording, a small craniotomy (~200 μm diameter) was made over the barrel cortex (*O'Connor et al., 2010*). The dura was left intact. Recordings targeting cortical L4 were obtained with patch pipettes pulled from borosilicate tubing (Sutter instrument, Novato, CA) and an Axopatch 700B amplifier (Molecular Devices, Sunnyvale, CA). Loose-seal juxtacellular pipettes were filled with ACSF or cortex buffer (in mM): 125 NaCl, 5 KCl, 10 dextrose, 10 HEPES, 2 CaCl$_2$, 2 MgSO$_4$, pH 7.4, osmolality ~272 mmol/kg. The manipulator depth was zeroed upon pipette tip contact with the dura. After contact, the craniotomy was covered by cortex buffer or 2% agar in cortex buffer. Aided by positive pressure (1 psi), the pipette was advanced through the dura. When searching for cells, the pipette pressure was reduced to 0.1–0.3 psi. Manipulator depths of 444–560 μm were considered L4 (*Andrew Hires et al., 2015*). Data acquisition was controlled by Ephus (*Suter et al., 2010*). The sampling rate was 10 kHz.

## Trial structure during illusory wall tracking

Trials were 200 cm long. Mice were rewarded with water on every trial irrespective of the angle error. For three mice, trials with turns at ± 11.3° and 0° were interleaved. For five mice, turns at ± 5.7°, ± 11.3°, 16.7°, and 0° were interleaved. Trials with no photostimulation, photostimulation of barrel cortex, and photostimulation of parietal cortex (three mice), and visual cortex (five mice) were interleaved. All mice were acclimatized to head-fixed running on the ball for 2–3 daily sessions, 15–30 min each before experiments began. Four mice were first tested on the wall-tracking task with real walls, and four mice were first tested with wall-tracking on the illusory walls.

## Trial structure during unilateral activation experiments

Photoactivation of L4 neurons occurred during an otherwise straight corridor. Left hemisphere and right hemisphere photostimulation trials were interleaved with trials without photostimulation. For three of the mice undergoing activation experiments, parietal cortex, 2.0 mm posterior to Bregma and 1.7 mm lateral to Bregma (*Harvey et al., 2012*), and barrel cortex were activated on interleaved trials. For the remaining five mice visual cortex and barrel cortex were activated on interleaved trials, as visual cortex had greater expression of ChR2 than parietal cortex. For three of the mice during activation experiments trials were only 100 cm long (with activation during the middle 50 cm) instead of 200 cm long (with activation during the middle 100 cm). Wall distance bias was calculated as the difference between the wall distance at the end of the trial and the mean wall distance on un-stimulated trials. For creating summary plots, data from manipulations of the left hemisphere were flipped in sign and averaged with data from manipulations of the right hemisphere.

## Cranial window surgery

Cranial window surgery was performed as described (*Huber et al., 2012*). A 3-mm diameter craniotomy was made over S1, centred at 1.5 mm posterior 3.4 mm lateral to Bregma in the left hemisphere. GCaMP6s virus was injected over a grid of 9 sites with 300 μm spacing between the sites, 20 nl per site. Injections were performed at a rate of 20 nl/80 s. Virus was injected with a volumetric injection system (Narishige, East Meadow, NY, MO-10 manipulator) and a bevelled pipette (20–30 μm inner diameter, Drummond Scientific, Broomall, PA; Wiretrol II Capillary; P/N 5-000-2010). The imaging window was constructed from two glass circles (150 μm thickness each). An inner circle, 3 mm diameter (Warner Instruments, Hamden, CT, 64-0720), and an outer circle, 5 mm diameter (Warner Instruments, 64-0700), were glued together with curable optical glue (NOR-61, Norland, Cranbury, NJ). The window was lowered into the craniotomy such that the outer circle rested on the bone and the inner circle rested on the brain. The space between the glass and the bone was filled with a layer of agar (2%). The window was secured in place with dental acrylic (Lang Dental, Wheeling, IL). Imaging sessions started 2–3 weeks after viral injection.

## Intrinsic optical imaging

Intrinsic optical imaging was performed through the cranial window as previously described (*O'Connor et al., 2010*). Mice were lightly anesthetized with isoflurane (0.5% ) and were placed on

a heat blanket with their bodies maintained at 37°. Images were acquired using a CCD camera (Retiga-2000RV, Qimaging, Surrey, BC, Canada) through a Leica MZ12.5 microscope (field of view 4.8 x 3.6 mm) with 630 nm LED illumination (Philips LumiLEDs, Amsterdam, Netherlands). The targeted whisker was placed inside a glass pipette connected to a piezoelectric bimorph. The whisker was deflected at 10 Hz for 4 s every 20 s for 10 min. Images were averaged during the 4 s stimulation epoch. A baseline image of the average of the 10 s period proceeding each stimulation epoch was subtracted of this image to generate the intrinsic signal image. Barrels were visible as regions showing decreased 630 nm reflectance. An image of the vasculature was taken with 530 nm LED illumination (Philips LumiLEDs) as a reference for alignment.

## 2-photon calcium imaging

The design of the 2-photon imaging system used has been described elsewhere (*Peron et al., 2015*). A Ti-Sapphire laser (MaiTai-HP, Spectra Physics, Irvine, CA) tuned to 1000 nm was used for excitation. Photons were detected using GaAsP photomultiplier tubes (10770PB-40). Imaging was performed through a 16x0.8 NA microscope objective (Nikon). The objective was moved by a piezo (PI) in the z-axis to enable multi-plane imaging. The beam was deflected along the x-axis by a resonant scan mirror (Thorlabs) to enable fast imaging. The field of view was 600 x 600 μm (512 x 512 pixels) over 3 planes separated by 20 μm in the z-axis, imaged at 7.8 Hz per plane. The top image was 100–150 μm below the pia. The microscope was controlled with Scanimage4 (https://openwiki.janelia.org/wiki/display/ephus/ScanImage). Average imaging power was < 40 mW, measured at the back aperture of the objective. GCaMP6s has a rise time of 200 ms and a decay time of 600 ms, which means the fast dynamics of neurons will be hard to capture during a calcium imaging experiment (*Chen et al., 2013*).

Imaging was performed during the wall-tracking task in closed-loop trials, in which the wall motion was coupled to the ball motion, interleaved with open-loop trials, in which the walls were moved to a fixed position independent of ball motion. Only data from open-loop trials were analysed. Open-loop trials lasted 8 s, and consisted of 2 s while the right side wall (contralateral to the imaging window) moved into place and the left side wall (ipsilateral to the imaging window) moved far out of reach, a middle 4 s when the right wall did not move, and 2 s while the walls returned to their centered positions, 19 mm from the face. The wall was allowed to come to the face not closer than 4 mm.

## Imaging analysis

Analysis of the calcium imaging data was performed using the Thunder library for neural data analysis written in Spark's Python API (*Freeman et al., 2014*) and in Python using the numpy, scipy, and pandas packages. To correct for brain motion, each image frame was registered to a reference image using global shifts obtained from the peak of an FFT based cross-correlation. The fluorescent time series of each pixel was regressed against the distance of the contralateral wall during open-loop trials when the mouse was running at least 3 cm/s. The wall distance and running speed of the mouse were down-sampled to the sampling rate to the imaging based on their mean values.

Regions of interest (ROIs) were drawn manually based on neuronal shape in an image of mean fluorescence across the session and an image of the local correlations of each pixel with a 5-pixel neighborhood (*Cheng et al., 2011*). Drawing ROIs without an activity independent anatomical marker (*Peron et al., 2015*) may lead to a slight bias towards active neurons. Baseline fluorescence, $F_0$, was determined using the 20$^{th}$ percentile of fluorescence in a 160-s rolling window. $\Delta F/F$ was computed as $(F–F_0 /F_0)$. Neuropil $\Delta F/F$ was calculated in a doughnut 3–8 pixels from the boundary of the ROI using the same procedure and subtracted from the ROI $\Delta F/F$ to generate the corrected $\Delta F/F$ signal. An ROI was taken to be active if the 90$^{th}$ quantile of its $\Delta F/F$ was greater than 1.0 $\Delta F/F$. Under similar conditions a single yields a $\Delta F/F$ of 0.3, with a half decay time of ~600 ms (*Chen et al., 2013*), suggesting that a 1.0 $\Delta F/F$ threshold corresponds to a train of 3–4 spikes.

For each ROI, on each trial a wall distance tuning curve was created by taking the average of the $\Delta F/F$ inside 3 mm bins of wall distance. The wall distance tuning curve was then created by averaging together these tuning curves on trials where the average speed of the mouse was greater than 3 cm/s. The significance of the tuning was assessed using an ANOVA. An ROI was taken to be tuned if

the p value of the ANOVA was less than 0.05 and the range of its tuning curve was larger than 0.3 ΔF/F. Tuning curves were smoothed with a univariate spline and a smoothing factor of 1.

## Electrophysiology

For silicon probe recordings, methods were similar to those previously described (*Guo et al., 2014*). Extracellular signals were recorded with A32-edge probes (A1x32-Edge-5 mm-20-177-A32, Neuronexus) that were connected to custom headstages (Intan Technology, Los Angeles, CA, fabricated at Janelia Research Campus, Brian Barbarits, Tim Harris, http://www.janelia.org/lab/apig-harris-lab). These headstages multiplexed the 32-channel voltage recording into 2 analog signals that were recorded on a PCI6133 board at 312.5 kHz (National Instrument, Austin, TX) and digitized at 14 bit. The signals were demultiplexed into the 32 voltage traces at the sampling frequency of 19531.25 Hz and saved and displayed using custom software spikeGL (C. Culianu, Anthony Leonardo, Janelia Research Campus). The headstage was connected to a motorized micromanipulator (MP-285, Sutter Instrument).

A small craniotomy (approximately 0.2 mm in diameter) was made over the C-row of the barrel field in mice already implanted with the clear-skull cap and headpost. The dura was left intact. The C-row barrel field was targeted using fluorescence from layer 4 of the barrel field in the *Scnn1a*-Tg3-Cre x Ai32 mice (*Figure 4—figure supplement 1a*). The dental acrylic and bone were thinned using a dental drill. The remaining thinned bone was carefully removed using a bent forceps. A separate craniotomy (~0.2 mm in diameter) was made through the headpost for ground wire in the cerebellum. A 2-mm-long platinum iridium ground wire (A-M Systems, Carlsborg, WA: 776000) was used as a ground wire.

To identify the position of the silicon probe in the brain, the top several millimeters of the probe were covered with DiI solution leaving electrode area clean. The probe was lowered 500 µm-700 µm deep into the cortex at rate of 20 µm per minute. The probe was rotated to 40° angle around the anterior posterior axis and inserted approximately perpendicular to cortical surface. The probe was then given around 15 min to settle before recordings started. At the end of every experiment layer 4 was photoactivated at 10 Hz with 2 ms pulses (1 mW-2 mW peak) to obtain current source density (CSD) traces as shown in *Figure 4—figure supplement 1b*. At the end of the experiment electrolytic lesions were made using the top and bottom pads of the silicon probe by passing 10 µA of current for 500 ms (Digitimer, Welwyn Garden City, UK: DS3). Immediately after the experiment mice were deeply anesthetized with 5% isoflurane then perfused with 0.1 M sodium phosphate buffer followed by 4% paraformaldehyde (PFA, in 0.1 M phosphate buffer, pH 7.4). The brain was immersed in fixative for at least 24 hr before sectioning.

Before sectioning, fluorescent images of the whole brain removed from the skull were taken with a microscope (Olympus, Tokyo, Japan: MVX10) to visualize the location of the silicon probe within the barrel field as shown in *Figure 4—figure supplement 1a*. The brain was then sectioned into 70-µm-thick coronal sections to allow for the visualization of the electrolytic lesions and the DiI stain tracks (*Figure 4—figure supplement 1b*). Combination of top and bottom lesions, the location of DiI traces, as well as the CSD analysis were critical for localization of the laminar position of the recording electrodes (*Figure 4—figure supplement 1b*). The pronounced minimum in the CSD trace was used to identify the middle of layer 4 for each recording that was used as a reference for all depth measurements. Each recorded unit then was assigned a depth value relative to L4 by extrapolating corresponding spike waveform amplitudes across the neighboring channels.

Mice were acclimatized to head-fixation and running on the ball for one to three days before the recording day. Each mouse was recorded for only one 2–4 hr long session, in a random sequence of open-loop and closed-loop experiments. Open-loop experiments were four seconds long and consisted of one second of the wall moving in towards a fixed distance from the mouse and one second of the wall moving away from the animal. The wall was allowed to come to the face not closer than 4 mm.

## Electrophysiology analysis

Raw voltages were band-pass filtered between 300 and 6000 Hz (MATLAB, Natick, MA: 'idealfilter'). Common noise was estimated by averaging across channels at each time-point (MATLAB 'trimmean', using 50% of the channels) and was subtracted from each of the individual channels. All

events that exceeded an amplitude threshold of three standard deviations of the background activity were detected. To avoid detecting multiple copies of synchronously spiking neighboring neurons threshold crossings that occurred within a 450 µs shadow period were merged within a mask covering several neighboring channels. The mask depth was adjusted dynamically and varied from 4 to 7 electrodes depending on neighboring activity. The event amplitudes across all 32 channels along with the first 3 PCAs of the waveform were used as features for spike sorting. Spikes were manually sorted in the klusters environment (http://neurosuite.sourceforge.net/). A waveform signal to noise ratio > 6 and the false alarm rate < 1.5% were used as quality control metrics (*Figure 4—figure supplement 1f*) (*Guo et al., 2014*). With these parameters 95% of the sorted units had a maximum spike rate > 1 Hz (*Figure 4—figure supplement 2c*).

All subsequent data analysis was done in Python using the numpy, scipy, and pandas packages. Spike times were binned in 2-ms bins and compared to the wall distance using the same methods as the calcium imaging. Analysis code and data can be found at https://github.com/sofroniewn/tactile-coding.

## Acknowledgements

We thank Bryan MacLennan and Simon Peron for help with cranial window surgeries and imaging; Brian Barbarits, Diego Gutnisky, Zengcai Guo, and Nuo Li for help with setting up electrophysiology and optogenetics; Jeremy Cohen, Nuo Li, Diego Gutnisky, Dan O'Connor, Kayvon Daie, Simon Peron, Hidehiko Inagaki, Kaspar Podgorski, Dario Campagner for comments on the manuscript.

## Additional information

### Funding

| Funder | Author |
|---|---|
| Howard Hughes Medical Institute | Nicholas James Sofroniew<br>Yurii A Vlasov<br>Samuel Andrew Hires<br>Jeremy Freeman<br>Karel Svoboda |
| International Business Machines Corporation | Yurii A Vlasov |

The funders had no role in study design, data collection and interpretation, or the decision to submit the work for publication.

### Author contributions

NJS, Optogenetic experiments, Imaging experiments, Electrophysiology experiments, Conception and design, Acquisition of data, Analysis and interpretation of data, Drafting or revising the article; YAV, Design and execution of electrophysiology experiments, Conception and design, Acquisition of data, Analysis and interpretation of data, Drafting or revising the article; SAH, Electrophysiological calibration experiments, Acquisition of data, Drafting or revising the article; JF, Analysis and interpretation of data, Drafting or revising the article; KS, Conception and design, Analysis and interpretation of data, Drafting or revising the article

### Author ORCIDs

Yurii A Vlasov, http://orcid.org/0000-0002-5864-3346

### Ethics

Animal experimentation: All procedures were in accordance with protocols approved by the Janelia Institutional Animal Care and Use Committee. (IACUC 14-115)

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
