## [Decision Letter]

Thank you for submitting your work entitled "Neural coding in barrel cortex during whisker-guided locomotion" for consideration by *eLife*. Your article has been favorably evaluated by Eve Marder (Senior Editor) and three reviewers, one of whom, Sacha Nelson, is a member of our Board of Reviewing Editors. One of the reviewers, Daniel Feldman, has also agreed to share his identity.

The reviewers have discussed the reviews with one another and the Reviewing Editor has drafted this decision to help you prepare a revised submission.

Summary:

This study investigates whisker-based sensory coding of wall distance in somatosensory cortex during innate wall following behavior in mice. This is an important extension of prior whisker coding studies in mouse S1, which have focused on simple detection, front-back localization, and surface properties. Using optogenetics, the authors show that S1 firing is necessary and sufficient for accurate wall following. Multi electrode recording, and calcium imaging, show that many S1 neurons encode wall distance, with most neurons preferring close distances, but many showing clear tuning for intermediate wall distances. In addition, many S1 neurons also encode direction of wall motion. Overall, these findings reveal a critical role for S1 activity in encoding distance estimation during wall following behavior.

Essentials revisions:

1) Although the text refers to layers 2/3 and 5, the figures do not make clear where the boundaries of L4 are. It would be clearer to actually assign units to specific laminae and not simply distance from the center of L4. If this is not possible, perhaps the average boundaries could be indicated in Figure 4—figure supplement 2.

2) Although there are other indications that activity in L2/3 is sparse, the authors should say more about a) whether or not there is overall agreement in the activity of cells studied electrophysiologically and with imaging and b) whether there were indications that the inactive cells were completely inactive or fired occasionally. Related to this is the question of how ROIs were chosen--i.e. is there any approximate sense of how the criteria stated in the methods translates into firing rate? Did a similar minimum firing rate apply in practice to units which could be isolated electrophysiologically?

3) It might be useful to include some additional analyses of the relationship between firing and running rate. It is not clear how tuning for wall distance and speed interact.

Additional suggestions:

One of the reviewers made the following suggestions for additional work that would add to the impact of the paper, but the reviewers agreed this would not be necessary for acceptance of the paper. Perhaps some of these issues could be addressed in the Discussion.

1) How is distance information read out from S1? The L4-ChR2 results argue for a simple firing rate code for proximity. However, the physiology shows many neurons that are tuned for intermediate distances, and the Discussion argues that such Gaussian tuning curves are metabolically more efficient than monotonic distance-response functions. If distance information is encoded by Gaussian-tuned neurons, why does ChR2 stimulation at varying intensities-which implements a monotonic population firing rate code-effectively drive wall-following behavior? (This fictive wall following is even more accurate than real-wall following!)

2) How is distance tuning generated for individual S1 neurons? Perhaps the simplest model is that each neuron's preferred distance matches the length of its principal whisker. This would make sense because more distant positions cannot directly drive spikes from the cell, while closer wall positions likely suppress firing by activating additional, shorter surround whiskers, which inhibit the recorded neuron. Such net inhibition between neighboring whiskers has been observed often in passive studies.

---

## [Author Response]

*Essentials revisions: 1) Although the text refers to layers 2/3 and 5, the figures do not make clear where the boundaries of L4 are. It would be clearer to actually assign units to specific laminae and not simply distance from the center of L4. If this is not possible, perhaps the average boundaries could be indicated in Figure 4—figure supplement 2.*

All the units had layer assignments, based on data similar to that presented in Figure 4—figure supplement 1. Units in different layers are now depicted with different grey scale values (new Figure 4—figure supplement 2 and Figure 4).

*2) Although there are other indications that activity in L2/3 is sparse, the authors should say more about a) whether or not there is overall agreement in the activity of cells studied electrophysiologically and with imaging and b) whether there were indications that the inactive cells were completely inactive or fired occasionally. Related to this is the question of how ROIs were chosen--i.e. is there any approximate sense of how the criteria stated in the methods translates into firing rate? Did a similar minimum firing rate apply in practice to units which could be isolated electrophysiologically?*

We added additional discussion comparing the electrophysiology and imaging data. We also moved some text about the sparseness of L2/3 activity in the calcium imaging data from the Methods to the Discussion section too.

We also added information on how regions of interest (ROIs) were selected and the expected firing rates for neurons judged as active based on calcium imaging.

We added information about the minimum maximum spike rate of the sorted spikes in the electrophysiology data to the methods (Figure 4—figure supplement 2).

*3) It might be useful to include some additional analyses of the relationship between firing and running rate. It is not clear how tuning for wall distance and speed interact.*

We now show additional analyses comparing tuning curves at slow fast speeds. The tuning curves are similar on average (new Figure 4—figure supplement 4). This topic is now discussed in the Results.

*Additional suggestions:*

*One of the reviewers made the following suggestions for additional work that would add to the impact of the paper, but the reviewers agreed this would not be necessary for acceptance of the paper. Perhaps some of these issues could be addressed in the Discussion. 1) How is distance information read out from S1? The L4-ChR2 results argue for a simple firing rate code for proximity. However, the physiology shows many neurons that are tuned for intermediate distances, and the Discussion argues that such Gaussian tuning curves are metabolically more efficient than monotonic distance-response functions. If distance information is encoded by Gaussian-tuned neurons, why does ChR2 stimulation at varying intensities-which implements a monotonic population firing rate code-effectively drive wall-following behavior? (This fictive wall following is even more accurate than real-wall following!).*

This is an interesting issue. The accuracy of tracking in the fictive corridor may depend on the details of the photostimulus (e.g. photostimulation frequency). Indeed, we found that the imposed change in activity (i.e. the spatial gradient of activity) with wall-distance is related to accuracy of tracking in the fictive corridor. We have added text to the Discussion to address this point.

*2) How is distance tuning generated for individual S1 neurons? Perhaps the simplest model is that each neuron's preferred distance matches the length of its principal whisker. This would make sense because more distant positions cannot directly drive spikes from the cell, while closer wall positions likely suppress firing by activating additional, shorter surround whiskers, which inhibit the recorded neuron. Such net inhibition between neighboring whiskers has been observed often in passive studies.*

We agree that this is a reasonable hypothesis for the non-monotonic wall distance tuning curves. We now discuss this possibility in the Discussion section.